

# A dataset of 30-meter annual vegetation phenology indicators (1985-2015) in urban areas of the conterminous United States

Xuecao Li[1], Yuyu Zhou[*1], Lin Meng[1], Ghassem R. Asrar[2], Chaoqun Lu[3], Qiusheng Wu[4]

[1]Department of Geological and Atmospheric Sciences, Iowa State University, Ames, IA, 50011, USA

[2]Joint Global Change Research Institute, Pacific Northwest National Lab, College Park, MD, 20740, USA

[3]Department of Ecology, Evolution, and Organismal Biology, Iowa State University, Ames, IA, 50011, USA

[4]Department of Geography, Binghamton University, SUNY, Binghamton, NY, 13902, USA

*Correspondence to*: Yuyu Zhou (yuyuzhou@iastate.edu)

**Abstract.** Fine-resolution satellite observations show great potential for characterizing seasonal and annual dynamics of vegetation phenology in urban domains, from local to regional and global scales. However, most previous studies were conducted using coarse or moderate resolution data, which are inadequate for characterizing the spatiotemporal dynamics of vegetation phenology in urban domains. In this study, we produced an annual vegetation phenology dataset in urban ecosystems for the conterminous United States (US), using all available Landsat images on the Google Earth Engine (GEE)

platform. First, we characterized the long-term mean seasonal pattern of phenology indicators of the start of season (SOS) and the end of season (EOS), using a double logistic model. Then, we identified the annual variability of these two phenology indicators by measuring the difference of dates when the vegetation index in a specific year reaches the same magnitude as its long-term mean. The derived phenology indicators agree well with *in-situ* observations from PhenoCam network and Harvard Forest. Comparing with results derived from the moderate resolution imaging spectroradiometer (MODIS) data, our Landsat

derived phenology indicators can provide more spatial details. Also, temporal trends of phenology indicators (e.g., SOS) derived from Landsat and MODIS are consistent overall, but the Landsat derived results from 1985 have a longer temporal span compared to MODIS from 2001. In general, there is a spatially explicit pattern of phenology indicators from the North to the South in cities in the conterminous US, with an overall advanced SOS in the past three decades. The derived phenology



product in the US urban domains at the national level is of great use for urban ecology studies for its fine spatial resolution (30 m) and long temporal span (30 years). The data are available at https://doi.org/10.6084/m9.figshare.7685645.v2.

## 1 Introduction

Dynamics of vegetation phenology in urban ecosystems play an important role in influencing human activities such as energy
use and public health. The change of vegetation greening and dormancy affects various ecological and environmental processes, such as carbon storage, energy use, water cycle, and climate change (Zhou et al., 2016;Keenan et al., 2014;Peng et al., 2013;Tang et al., 2016). These influences are amplified in urban ecosystems due to the notably altered urban environment by anthropogenic activities. For example, the urban heat island (UHI) results in an earlier start and a longer duration of the growing season than surrounding rural areas (White et al., 2002;Zhang et al., 2004b;Jochner et al., 2011). The change of
vegetation phenology affects the start and duration of pollen season in urban domains, which has become a major concern by public health authorities for the potential risks of pollen-induced respiratory allergies (e.g., asthma) (Aas et al., 1997;Anenberg et al., 2017;Gong et al., 2012;Li et al., 2019b). Furthermore, the rapid pace of urbanization is expected to continue in the future, with more than 66% of the world's population residing in urban areas by 2050 (United Nations, 2018), which will result in a more notable effect of urban environment change. However, our knowledge about the vegetation phenology response to
urbanization under different development scenarios is still unclear, partly because of the difficulties in observing and mapping the dynamics of vegetation phenology at fine spatial and temporal resolutions in/around urban areas. Therefore, dynamics of vegetation phenology in urban domains is crucial for understanding the response of vegetation phenology to urbanization, and this further helps to develop process-based phenology models for prediction under the compound effect of global warming and urbanization (Jochner and Menzel, 2015;Jochner et al., 2011;White et al., 2002).

Medium and coarse resolution satellite observations are inadequate to support vegetation phenology studies in urban domains, although they have been extensively used for phenology mapping. Relevant studies include using the advanced very high-resolution radiometer (AVHRR) data (Moody and Johnson, 2001;White et al., 2002;Piao et al., 2006;Cong et al., 2012), and the moderate resolution imaging spectroradiometer (MODIS) data (Zhang et al., 2004b;Zhou et al., 2016;Walker et al., 2012;Walker et al., 2015;Liu et al., 2016). The primary advantage of these datasets is their long-term observations with a fine

temporal resolution. However, the relatively coarse (1-8 km) spatial resolution is limited to capture the spatial heterogeneity of phenology in urban domains (White et al., 2002;Hogda et al., 2002). In contrast, Landsat observations with a fine spatial resolution (30 m) and a long temporal span (since the 1980s) offer the opportunity to overcome this limitation (Zipper et al., 2016;Li et al., 2017b).

There are few attempts of mapping vegetation phenology in urban domains using Landsat observations at a large scale due to complex vegetation compositions in urban ecosystems and the huge dataset required for analysis. Despite the high spatial resolution and long-term record of Landsat, the 16-days revisit frequency and the cloud coverage make it difficult to collect adequate observations to composite the time series of vegetation indices for investigating vegetation phenology dynamics. Therefore, the long-term mean pattern of vegetation phenology using multi-year observations were generally investigated in

most Landsat-based phenology studies. After that, the annual variability of phenology indicators can be identified through measuring the difference of dates when the vegetation index in a specific year reaches the same magnitude as its long-term mean (Fisher et al., 2006;Melaas et al., 2013). However, currently this approach was mainly used in natural ecosystems (e.g., deciduous forest) or at local scales (Fisher et al., 2006;Melaas et al., 2013;Li et al., 2017b), and there lack large-scale applications in urban domains. First, vegetation types and compositions in urban ecosystems are more complicated. The

seasonal pattern of vegetation growth varies among different vegetation types, which requires a more generalized approach to filter out available Landsat observations for a specific year to measure its gap to the long-term mean (Li et al., 2017b). Second, an improved understanding of vegetation phenology in urban areas over different regions require massive Landsat observations and super-computational power. More than one thousand Landsat scenes need to process for mapping vegetation phenology dynamics in a given city, and this number is huge when expanding the mapping area at the national or global scales.

The advent of Google Earth Engine (GEE) platform provides the possibility to map vegetation phenology dynamics using the long-term Landsat data at the regional and global scales. GEE is a start-of-art platform for planetary-scale data analysis, mapping, and modelling, owing to free access to numerous global datasets and advanced computational capabilities (Gorelick et al., 2017). There are several successful studies built on the GEE platform for mapping long-term dynamics of forest and water, using all available Landsat images at the global scale (Hansen et al., 2013;Pekel et al., 2016). It is convenient to

composite time series data of vegetation index at the pixel level on the GEE, using all clear-sky pixels. Also, the capability of

cloud-based computation offered by the GEE enables efficient and effective mapping practices at different spatial and temporal scales (Xiong et al., 2017).

To better support vegetation phenology studies in urban domains with more details, for the first time, we mapped annual phenology (1985-2015) using long-term Landsat observations at a high spatial resolution in the US and characterized the

dynamics of urban vegetation phenology. The remainder of this paper describes the study area and data (Section 2), the adopted method for mapping vegetation phenology indicators (Section 3), the results with discussion (Section 4), and concluding remarks (Section 5).

## 2 Study area and data

Our study area includes all cities greater than 500 km$^2$ and their surrounding rural areas in the conterminous US. First, the

urban extent was derived from nighttime light (NTL) observations (2013) (Zhou et al., 2018;Zhou et al., 2014). Then, a buffer zone with the same size as the urban area was identified as the surrounding rural area. The near equal size of urban and rural areas enables us to explore the response of vegetation phenology to urbanization through characterizing their phenology differences (Li et al., 2017a). In total, 148 urban clusters with different sizes were identified for deriving phenology indicators and their dynamics (Fig. S1).

Landsat surface reflectance data is our primary dataset used for vegetation phenology mapping. Images obtained from different sensors, i.e., Thematic Mapper (TM), Enhanced Thematic Mapper Plus (ETM+), and Operational Land Imager (OLI), were used to composite the time series of the enhanced vegetation index (EVI) of each pixel (Huete et al., 2002). The surface reflectance data have been corrected for the radiometric, topographic, and atmospheric effects (Masek et al., 2006). Clouds and shadows were removed before compositing the EVI time series. Thus, all available clear-sky pixels during 1985-2015

were used in our analysis.

## 3 Method

We developed an automatic approach to map urban vegetation phenology indicators using long-term (1985-2015) Landsat images on the GEE platform (Fig. 1). First, we composited the EVI time series using all clear-sky observations at the pixel



level, ordered by the day of year (DOY). A double logistic model was then applied on the derived EVI time series to obtain

the long-term mean pattern of phenology indicators (start of season (SOS) and end of season (EOS)) in Fig. 1a). Second, we

derived the annual variability of phenology indicators in urban and surrounding rural areas (Fig. 1b), by measuring the

difference of dates when the EVI in a specific year reaches the same magnitude as its long-term mean (Li et al., 2017b). Details

of each step are presented in the following sections.

### 3.1 Long-term mean phenology indicators

We composited EVI observations over the years to reflect its seasonal change before the implementation of the double logistic

model. First, we used all clear-sky observations of EVI and ordered them by their DOYs. This step allows us to retrieve the

seasonal pattern of vegetation dynamics using multi-year data because the temporal distribution of Landsat data is uneven due

to the satellite revisit time and sky conditions. Then, we applied a smoothing procedure using a moving average of continuous

observations within 2 days to minimize the impact of abnormal observations. This procedure can keep the raw seasonal pattern

of EVI (Fig. S2), and further helps to reduce the uncertainty of parameter estimation in the double logistic model.

We characterized the seasonal change of vegetation growth using a double logistic model. This model has several advantages

when compared to other approaches such as the splines and harmonic models (Melaas et al., 2016b;Carrão et al., 2010): (1) it

captures the green-up and senescence phases using different sigmoid functions; and (2) the physical meaning of parameters is

related to the vegetation growth and senescence (Fisher et al., 2006;Li et al., 2017b). The derived EVI time series data were

fitted using the double logistic model as Eq. 1.

$$f(t) = v_1 + v_2 \left( \frac{1}{1+e^{-m_1(t-n_1)}} - \frac{1}{1+e^{-m_2(t-n_2)}} \right)$$    (1)

where $f(t)$ is the fitted EVI value at the day $t$; $v_1$ and $v_2$ are the background and amplitude of EVI over the entire year,

respectively; and pair-parameters (i.e., $m_1$ & $n_1$, $m_2$ & $n_2$) capture the trends of green-up and senescence phases of vegetation

growth, respectively.

We developed a stepwise statistical approach to estimate the parameters of the double logistic model on the GEE platform for

large-scale applications because currently the GEE platform does not support for optimization of parameters. Calculation of

these parameters was presented in the Appendix. In general, the performance of this GEE-based double logistic model is robust

for different land cover types, and the derived results are close to that from the optimization algorithm (Fig. 2). For example,

although the magnitude of EVIs is relatively low in urban areas with low vegetation cover, a distinctive seasonal pattern of vegetation growth can be captured by the double logistic model. Also, sigmoid curves during green-up and senescence phrases are notably different across different vegetation cover types (e.g., forest and cropland). We evaluated the performance of fitted double logistic model based on the correlation between the fitted and observed EVI observations to identify pixels with land use/cover change during the study period or pixels with weak vegetation signals (e.g., purely built-up area or barren). This stepwise statistical approach can be implemented at the pixel level on the GEE platform in a parallel manner, which significantly improved our mapping efficiency at the large scale.

We derived phenology indicators of SOS and EOS using the half-maximum criterion (Fisher et al., 2006). Based on this criterion, SOS and EOS were defined as dates when the derivative of EVI reaches the maximum during the green-up and senescence phases. Although there are other definitions of SOS and EOS such as inflection points (Zhang et al., 2003), the criterion used in our study is advantageous because: (1) they represent the dates when most leaves are likely to emerge (i.e., the steepest points on the symmetric sigmoid curves); and (2) they are temporally more stable and can be applied to plants with different canopy structures (Fisher and Mustard, 2007). The growing season length (GSL) was defined as the difference between EOS and SOS.

### 3.2 Annual variability of phenology indicators

We derived the annual variability of vegetation phenology indicators using the developed generalized Landsat phenology (GLP) approach (Li et al., 2017b). Considering the temporally uneven distribution of available Landsat observations over the years, the annual variability of phenology indicators was measured as the difference of dates when the EVI in a specific year reaches the same magnitude as its long-term mean (Fisher et al., 2006;Melaas et al., 2013). Only EVI observations in the rational ranges of DOY and EVI (empty circles in shaded frames) in a given year were used in the GLP approach (Fig. 3). Observations outside this range (the shaded frames), which are either outliers or beyond the temporal ranges of green-up and senescence phases, were not used in calculating the annual variability. In the GLP approach, we also designed a self-adjusting strategy to derive the bounds of the shaded frames in the green-up and senescence phases (Fig. 3). For the green-up phase, the rational DOY ranges (two points on the long-term mean curve that intersected with the shaded green frame in Fig. 3) were defined as the dates when change rates (or derivative) of EVI reach the half-maximum before and after the date of SOS (i.e.,

the date with the maximum change rate). Thus, the corresponding EVI ranges were calculated based on the derived DOY

ranges and the long-term mean curve. The rational ranges for the senescence phase were determined using the similar approach.

This approach already showed its applications for different vegetation types (e.g., cropland or forest) with varying seasonal

patterns of EVI. More details about this approach can be found in Li et al. (2017b).

## 4 Results and discussion

### 4.1 Performance of the GEE-based double logistic model

The performance of developed GEE-based double logistic model is reasonably well across different latitudes and along the

urban-rural gradient. Take forest as an example, the seasonal pattern of EVI varies from the South to the North in the US, with

notably different sigmoid curves for the green-up and senescence phrases (Fig. 4). Our fitting approach can well capture the

diverse seasonal patterns of EVI for forest across space. Also, the developed approach shows the good capability of fitting

EVI time series along the urban-rural gradient (Fig. 5), where the vegetation composition and the seasonal pattern of EVI are

more complicated compared to natural ecosystems. For sites in urban center, although their EVIs are low, a distinctive seasonal

pattern with a good fitting was observed.

### 4.2 Comparison with PhenoCam data

The derived phenology indicators (SOS and EOS) are spatially consistent with *in-situ* PhenoCam data overall (Fig. 6).

PhenoCam is a regional-scale network of digital cameras that provide high temporal resolution vegetation canopy and

phenology information (Richardson et al., 2018). The records in PhenoCam are observed green chromatic coordinate (GCC),

which is used as the indicator of vegetation dynamics. We used all PhenoCam sites in the US and compared the mean SOS

and EOS with Landsat derived results. The definition of SOS and EOS we used in the PhenoCam data (i.e., the half-maximum

criterion) is consistent with our result derived from Landsat data. Overall, correlations of the derived SOS and EOS from the

Landsat and PhenoCam are 0.66 and 0.43, respectively. Most indicators are around the 1:1 line, indicating a close

correspondence of phenology indicators derived from these two independent datasets. For those sites (blue or light blue dots)

with large differences, the fitting of Landsat EVIs using the double logistic model is relatively worse. These sites are mainly

distributed in ecosystems dominated by shrubs, evergreen forests, or wetlands (Fig. S3). With correlation coefficients lower

than 0.85 (worse fitting) excluded as suggested by Melaas et al. (2016b), the overall agreements between Landsat and PhenoCam derived results were notably improved to 0.86 and 0.94, for SOS and EOS, respectively. Discrepancies between these two sets of indicators derived from Landsat and PhenoCam are mainly attributed to two factors including: (1) using two different vegetation indices (i.e., EVI and GCC) (e.g., relatively weak EVI but strong GCC for sites in arid regions with sparse

plants); and (2) the effect of field of view for *in-situ* PhenoCam and space-based Landsat observations (Liu et al., 2017).

The annual variability of phenology indicators (SOS and EOS) derived from Landsat observations also shows a good agreement with *in-situ* PhenoCam results (Fig. 7). We selected 11 deciduous broadleaf forest sites for comparison with continuous observations of more than five years (Fig. 7a). Landsat pixels located within 500 m of each PhenoCam station were used to ensure adequate samples to reflect the vegetation phenology dynamic at the local scale for this comparison (Melaas et

al., 2016a). The temporal dynamics of Landsat derived SOS and EOS generally follow the changes captured by PhenoCam observations (Fig. 7 b-c). A detailed illustration of the *Acadia* site indicates the SOS derived from the two datasets is notably decreasing during period 2006-2010 and their corresponding EOS is increasing after 2011. Although magnitudes of SOS and EOS are different over the years, their temporal trends (i.e., decreasing or increasing) are relatively consistent. Overall, the annual SOS indicator derived from Landsat shows a better agreement (0.74) with the result obtained from *in-situ* PhenoCam

observations (Fig. 7d). The agreement of annual variability of Landsat and PhenoCam EOS is relatively weaker (0.26) (Fig. 7e), which is consistent with previous results reported by Melaas et al. (2016a). The main reason for the weak agreement of annual variability of EOS is the difference in greening represented by GCC and EVI. That is, in the green-up phase, both GCC and EVI are rapidly increasing. While in the senescence phase, the EVI detected by Landsat slightly decreases, which is notably different from the pattern reflected by GCC that rapidly decreases once the leaf color changes.

**4.3 Comparison with Harvard Forest phenology data**

Our Landsat derived phenology indicators also show a good agreement with that from the Harvard Forest (HF) over the past decades (Fig. 8). The HF data were collected by field observers in spring and fall seasons for more than 25 years (Richardson et al., 2006). Our Landsat derived indicators were compared with dates of SOS and EOS recorded in the HF data, in which SOS and EOS are defined as the dates when the leaf length reaches 50% of its final size and the leaf color reaches 10% of the

color change to the greenest, respectively (Melaas et al., 2016a). Three dominant species of red oak (*Quercus rubra; QURU*),

red maple (*Acer rubrum; ACRU*), and yellow birch (*Betula alleghaniensis; BEAL*) in the HF were included in our analysis.

Overall, the SOS of the three-dominant species in the HF is similar and shows a good agreement with SOS derived from

Landsat observations (Fig. 8a). The RMSE between Landsat SOS and the HF data is 3.5 day, and the correlation coefficient is

0.81 (Fig. 8c), indicating a closer SOS and a relatively consistent temporal pattern. EOS shows a relatively larger gap among

species (Fig. 8b), i.e., the EOS of red oak is notably later compared to other two species of red maple and yellow birch. The

Landsat derived EOS is within the range of EOS of the three species, and the temporal variability of two data sources are

similar, although their magnitudes are different. The RMSE between Landsat EOS and the HF data is 3.7 day, and the

correlation coefficient is 0.51 (Fig. 8d).

**4.4 Comparison with MODIS data**

Phenology indicators (e.g., SOS) derived from Landsat observations provide more spatial details in/around urban areas and

are spatially consistent with those from MODIS (Fig. 9). Taking the Chicago metropolitan area as an example, we compared

the Landsat derived SOS with that from MODIS in two ways. First, we estimated SOS from the MODIS EVI (16-day) using

the same approach for Landsat. Second, we retrieved SOS from the widely used MODIS phenology product (MCD12Q2)

(Zhang et al., 2003). It is worthy to note that the SOS defined in MCD12Q2 is the inflection point of EVI growth during the

green-up phrase, and this definition is different from our half-maximum criterion (Fisher and Mustard, 2007). Therefore, the

SOS of MCD12Q2 is generally earlier than the other two. Also, there are uncertainties in MCD12Q2 in highly urbanized

regions, where the SOS is above 180 days (Fig. 9a). Overall, more spatial details of SOS can be revealed in results derived

from Landsat compared to MODIS (Fig. 9b). In highly urbanized regions, Landsat SOS can also capture the seasonal pattern

of vegetation growth. Normalized SOSs derived from MODIS and Landsat show a relatively consistent trend along the gradient

of developed areas (Fig. 9c), although their magnitudes are different (Fig. 9a).

Landsat derived phenology indicator of SOS exhibits a consistent temporal pattern compared to MODIS with a longer temporal

span (Fig. 10). Although the temporal distribution of Landsat is uneven compared to MODIS, the annual variability of

phenology indicators can be captured well using the clear EVI observations in a given year relative to the long-term mean

pattern. For example, there is a notable advancement of SOS in 2012, and all three SOSs captured this variability at the pixel

and regional levels (Fig. 10a and 10b). The magnitude difference of derived SOS between Landsat and MCD12Q2 is mainly

due to their definitions, and the difference of SOS between Landsat and MODIS EVI is likely caused by scale effect (e.g., mixed pixels).

**4.5 Spatiotemporal patterns of phenology indicators**

Phenology indicators (SOS, EOS, and GSL) in urban domains exhibit a spatially explicit pattern from the North to the South

in the conterminous US, with an overall advanced SOS in the past three decades (Fig. 11). SOS becomes earlier and EOS becomes later along the latitudinal gradient, although such spatial difference is more discernible in SOS compared to EOS at the national scale. As a result, GSL shows a generally extended trend from the North to the South (Fig. 11a). This spatial pattern of phenology indicators (e.g., SOS) is also confirmed at the city level with more details (Fig. 11b). Meanwhile, the SOS is advanced in the past three decades, particularly in cities in the northern US (e.g., Boston). Spatiotemporal patterns of

phenology indicators in the conterminous US reflect the response of vegetation phenology to regional differences of elevation, temperature, precipitation, vegetation type, as well as the global warming in past decades (Zhang et al., 2004a;Li et al., 2017a).

**5 Data availability**

The derived vegetation phenology data in urban domains are available at https://doi.org/10.6084/m9.figshare.7685645.v2 (Li et al., 2019a).

**6 Conclusions**

This study generated the first national-scale dynamics of annual vegetation phenology in urban domains (all cities greater than 500 km$^2$ and their surrounding rural areas) using long-term (1985-2015) Landsat observations on the GEE platform. First, we mapped the long-term mean seasonal pattern of vegetation dynamics using a double logistic model. In this step, we proposed a stepwise statistical approach to estimate parameters in the double logistic model and implemented it on the GEE platform.

Next, we identified annual dynamics of phenology indicators (i.e., SOS and EOS) by measuring the difference of dates when the EVI in a specific year reaches the same magnitude as its long-term mean. Finally, we developed the first high spatial resolution (30 m) phenology product in urban areas in the conterminous US, over past three decades (1985-2015).



The Landsat based phenology indicators show good agreements with those derived from independent *in-situ* observations (PhenoCam and HF) and another widely used satellite observations from MODIS. Overall, the phenology indicators derived from Landsat and PhenoCam are consistent for their long-term mean and annual variability. The comparison with field observations collected in the HF suggests the Landsat derived indicators can capture the temporal dynamics of vegetation

phenology in this forest ecosystem. Besides, the Landsat derived phenology indicators can provide more spatial details in/around urban areas, compared to the moderate-resolution MODIS results. Also, the temporal trends of phenology indicator (e.g., SOS) derived from Landsat and MODIS are consistent overall, and Landsat additionally extends the temporal span than MODIS back to the past three decades.

The Landsat phenology product in urban areas is of great use in urban phenology studies such as phenology response to

urbanization. There is a spatially explicit pattern of phenology indicators from the North to the South in US cities, with an overall advanced SOS in the past three decades. With this new phenology dataset (with a long temporal coverage and a high spatial resolution), the response of vegetation phenology to urbanization (e.g., UHI) can be further investigated, particularly for plants in the urban center or suburban areas with notably altered urban environment by anthropogenic activities, where most people reside (Zhang et al., 2004b;Alberti et al., 2017). This dataset, together with ground-based pollen concentration

data, is also of help in decision making relevant to pollen-induced allergy diseases (Li et al., 2019b).

**Appendix**

The double logistic model used in the GLP approach includes two sigmoid curves indicating the green-up and senescence phases of vegetation growth (Eq. A1).

$$f(t) = v_1 + v_2\left(\frac{1}{1+e^{-m_1(t-n_1)}} - \frac{1}{1+e^{-m_2(t-n_2)}}\right) \qquad \text{A1}$$

where $f(t)$ is the fitted EVI value at the day $t$; $v_1$ and $v_2$ are the background and amplitude of EVI over the entire year, respectively; the first sigmoid ($Sig_1: \frac{1}{1+e^{-m_1(t-n_1)}}$) with pair-parameters of $m_1$ & $n_1$ captures the green-up phase of vegetation growth; and the second sigmoid ($Sig_2: \frac{1}{1+e^{-m_2(t-n_2)}}$) with pair-parameters of $m_2$ & $n_2$ captures the senescence phase of vegetation growth (Fig. A1).





We derived six parameters (i.e., $v_1$, $v_2$, $m_1$, $n_1$, $m_2$, and $n_2$) in the double logistic model using a statistics approach on the

GEE platform. First, we estimated $v_1$ and $v_2$ based on the smoothed EVI time series, with abnormal observations (or noise)

excluded. We calculated the quantile levels of 5[th] and 95[th] as the minimum $v_1$ and maximum EVI $v_{max}$ over the entire DOY

range, to avoid possible biases caused by extreme values. Thus, $v_2$ can be determined as Eq. A2.

$$v_2 = v_{max} - v_1 \qquad\qquad\qquad \text{A2}$$

The first part ($Sig_1$) of the double logistic model in the green-up phase (Eq. A3) can be translated to Eq. A4 by using the

smoothed EVI time series only during the green-up phase before $doy_{max}$ and converted into a logarithmic form as Eq. A5.

$$Sig_1 = \frac{f(t) - v_1}{v_2} = \frac{1}{1 + e^{-m_1(t - n_1)}} \qquad\qquad \text{A3}$$

$$\frac{v_1 + v_2 - f(t)}{f(t) - v_1} = e^{-m_1(t - n_1)} \qquad\qquad \text{A4}$$

$$\ln\left(\frac{v_1 + v_2 - f(t)}{f(t) - v_1}\right) = -m_1(t - n_1) \qquad\qquad \text{A5}$$

where the left term in Eq. A5 can be calculated using $v_1$ and $v_2$, as well as the smoothed EVI time series $f(t)$ only during the

green-up phase before $doy_{max}$. $m_1$ and $n_1$ can be estimated using the least square regression approach.

Finally, based on the estimated parameters (i.e., $v_1$, $v_2$, $m_1$ and $n_1$), the second part ($Sig_2$) of the double logistic model in the

senescence phase can be formulated as Eqs. A6-8, respectively. In a similar manner, the pair-parameters of $m_2$ and $n_2$ can be

15   estimated using the least square regression approach, and the smoothed EVI time series during the green-up and senescence

phases together.

$$Sig_2 = \frac{v_1 + v_2 Sig_1 - f(t)}{v2} = \frac{1}{1 + e^{-m_2(t - n_2)}} \qquad\qquad \text{A6}$$

$$\frac{v_2(1 - Sig_1) - v_1 + f(t)}{v_1 + v_2 Sig_1 - f(t)} = e^{-m_2(t - n_2)} \qquad\qquad \text{A7}$$

$$\ln\left(\frac{v_2(1 - Sig_1) - v_1 + f(t)}{v_1 + v_2 Sig_1 - f(t)}\right) = -m_2(t - n_2) \qquad\qquad \text{A8}$$

20   **Author contributions**

ZY and LX designed the research; LX and ZY implemented the research and wrote the paper; GA, ML, LC, and WQ revised

the manuscript.



**Competing interests**

The authors declare that they have no conflict of interest.

**Acknowledgments**

This study was supported by the NASA ROSES INCA Program "NNH14ZDA001N-INCA".

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





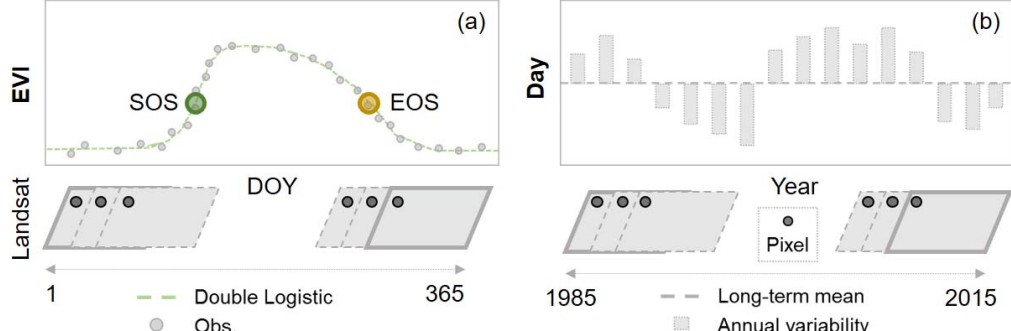

**Fig. 1:** The proposed framework for deriving long-term (1985-2015) mean vegetation phenology indicators (SOS and EOS) (a) and their

annual variabilities (b).




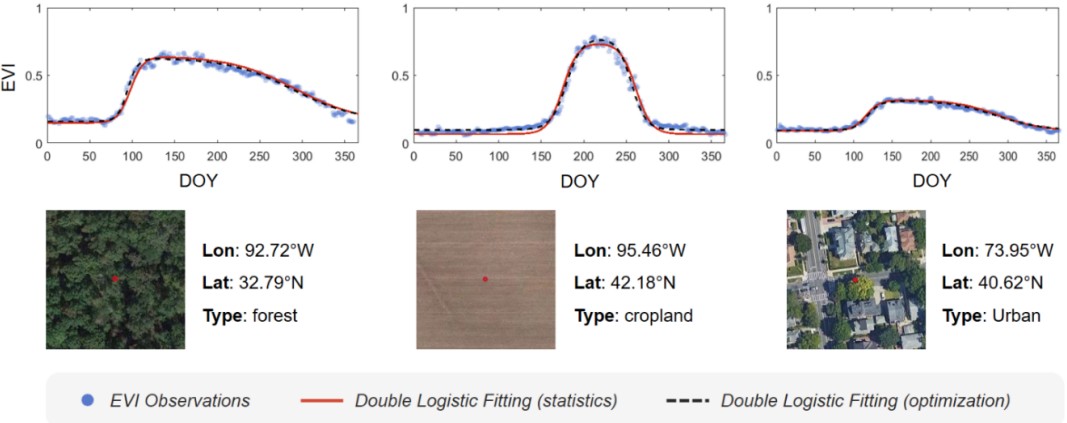

**Fig. 2:** Seasonal patterns of vegetation growth captured by the double logistic model for three distinctly different land cover types. The extent of a snapshot is 100m × 100m, and the red dot in the snapshot is the location of the EVI plot. EVI observations were composited

5    using all clear-sky pixels during the past three decades (1985-2015).





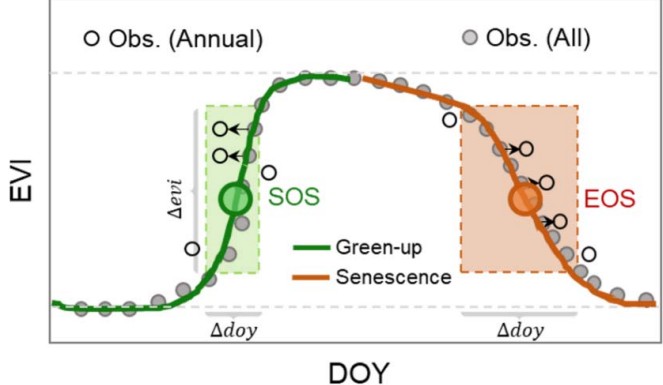

**Fig. 3:** Illustration of the GLP approach for identifying the annual variability of phenology indicators. The solid circles are long-term EVI observations and the empty circles are observations at a specific year. The shaded frames colored as green and brown are the rational ranges

5    of DOY and EVI to be used during the green-up and senescence phases, respectively.



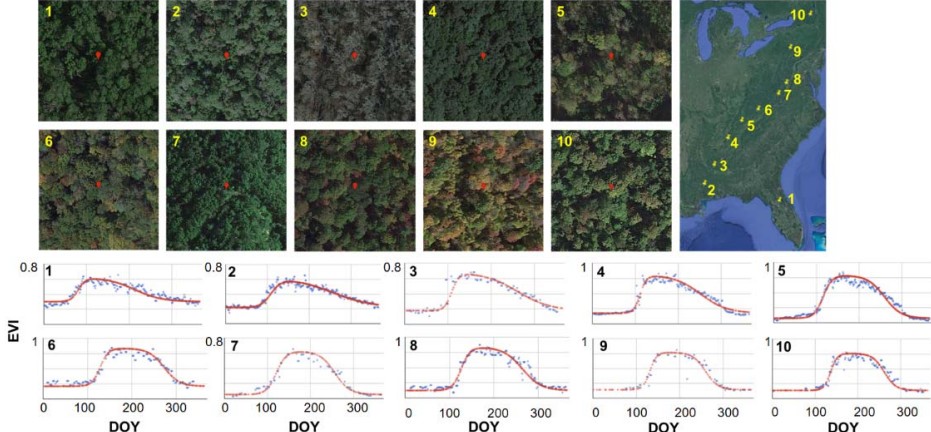

**Fig. 4:** Performance of the GEE-based double logistic model from the South to the North in the US using forest as an example. Each snapshot

indicates a 1km2 square, and the red dot in the middle is the location (30m) of EVI time series fitting.





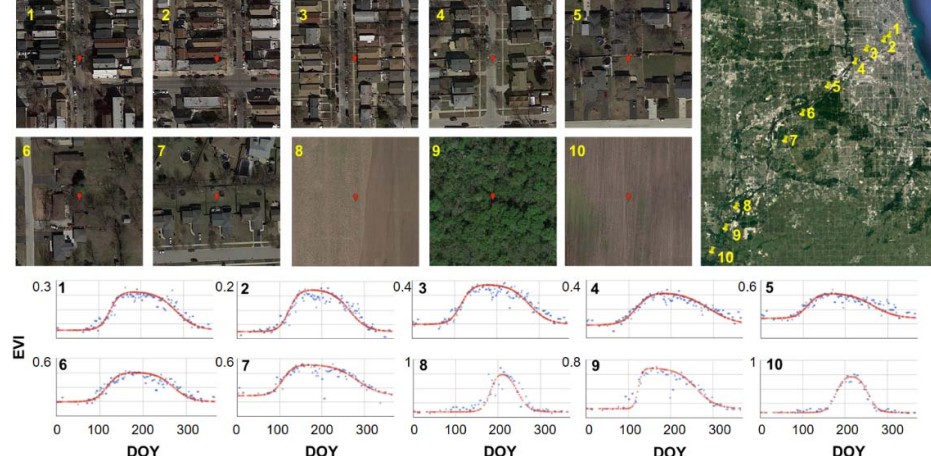

**Fig. 5:** Performance of the GEE-based double logistic model for sites along an example urban-rural gradient in the Chicago metropolitan

area. Each snapshot indicates a 1km2 square, and the red dot in the middle is the location (30m) of EVI time series fitting.





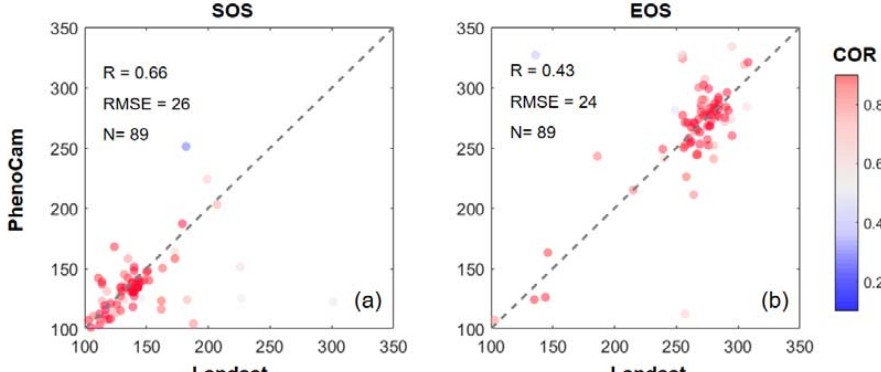

**Fig. 6:** Comparison of the period (2001-2015) mean phenology indicators of SOS (a) and EOS (b) derived from Landsat and PhenoCam

observations. COR: the correlation coefficient between the raw and fitted EVIs using the double logistic model.





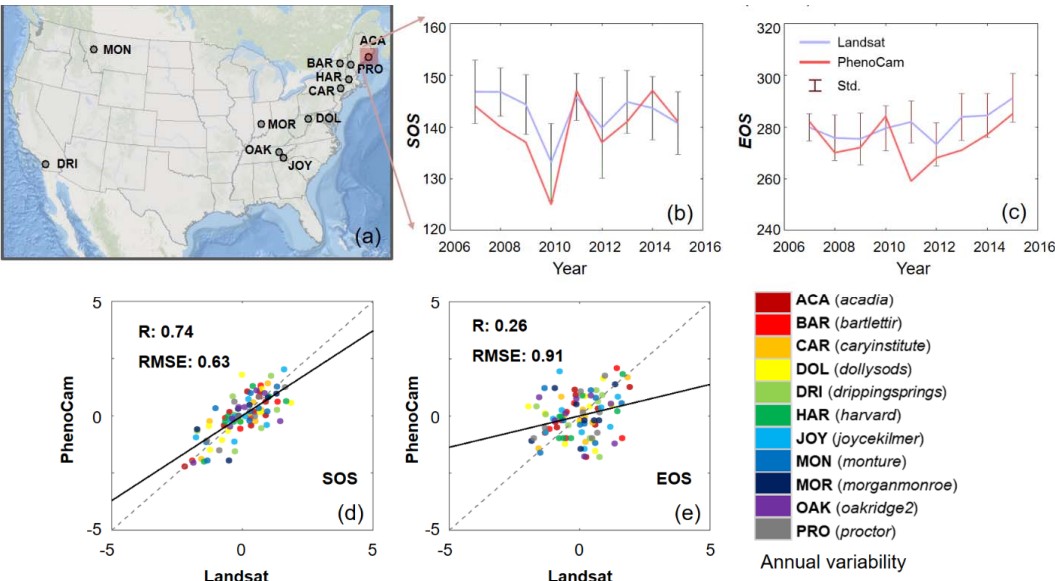

**Fig. 7:** Selected PhenoCam sites of deciduous broadleaf forest (a). Annual time series of phenology indicators in the station of *Acadia* for

SOS (b) and EOS (c). Comparison of annual variability of SOS (d) and EOS (e) between Landsat and PhenoCam phenology indicators

across all stations. The annual variability for each site is defined as $(x - \mu)/\sigma$, where $x$ is the annual value of SOS and EOS, $\mu$ and $\sigma$ are

5    mean and standard deviation of SOS or EOS over the years.



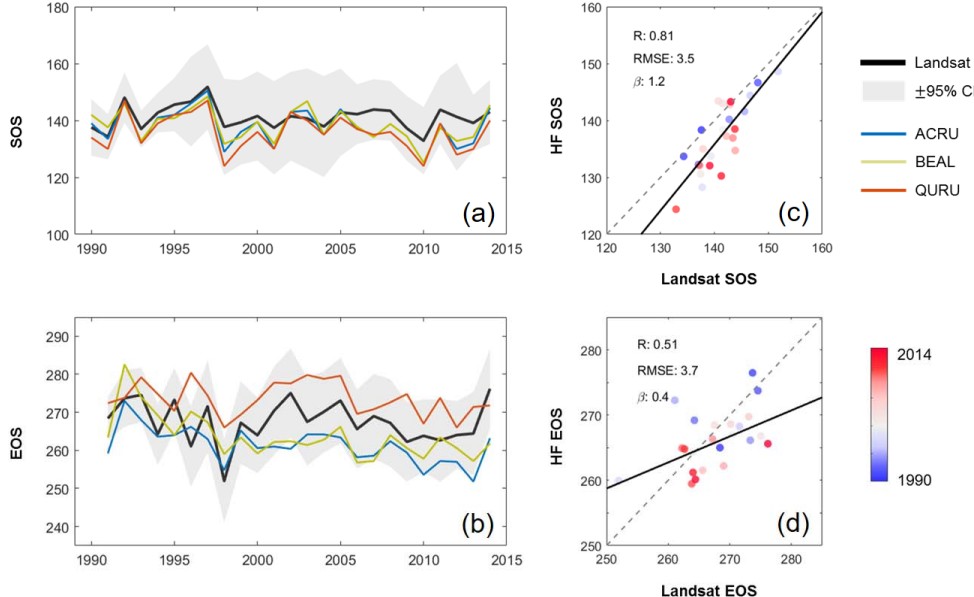

**Fig. 8:** Annual dynamics of SOS (a) and EOS (b) derived from Landsat and Harvard Forest (HF) observations and their scatter plots of SOS

(c) and EOS (d) over the years.





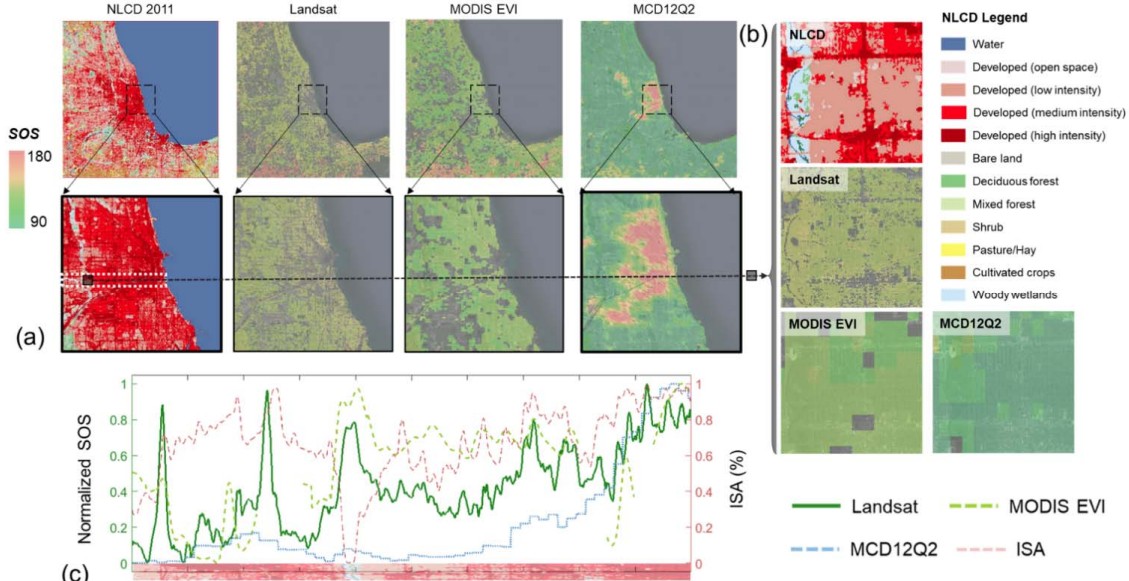

**Fig. 9:** Spatial patterns of the mean SOS (2001-2014) derived from Landsat, MODIS EVI, and MCD12Q2 and the land cover from the national land cover database (NLCD) (2011) in the Chicago metropolitan area (a). Enlarged views (b) at the location of the black square in

5    (a). Change of normalized SOS and impervious surface area (ISA) (c) along the white rectangle in (a) (from left to right). Pixels without good fitting performance (i.e., the correlation coefficient is lower than 0.85) were removed in the derived SOS from Landsat and MODIS EVI.





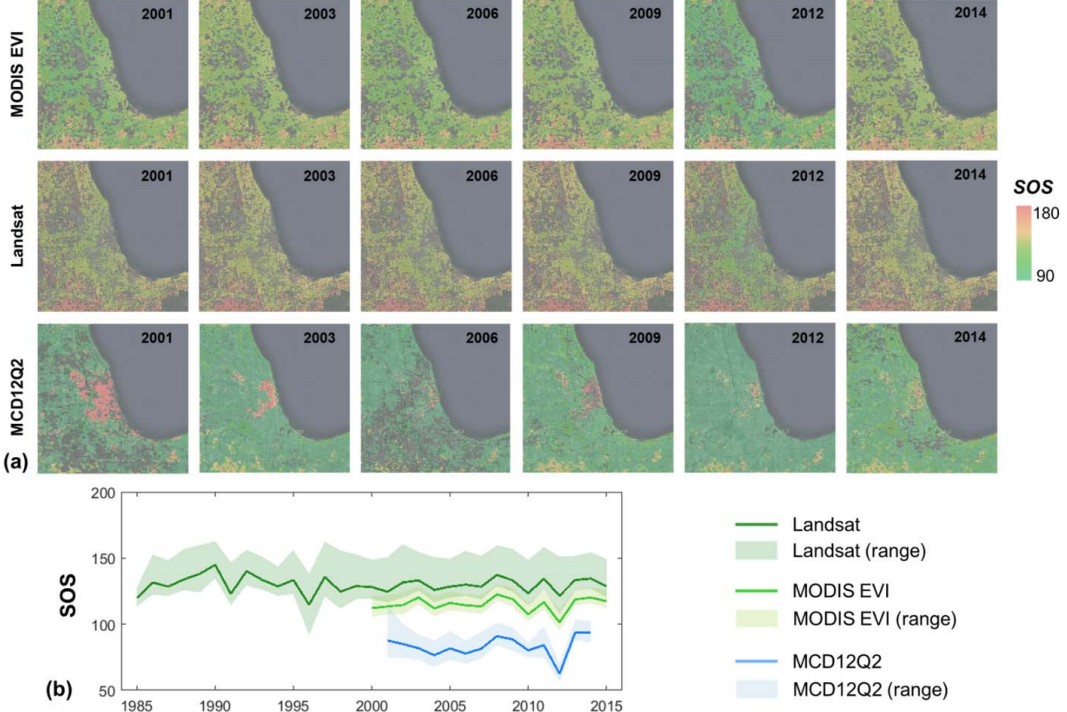

**Fig. 10:** Annual SOS derived from Landsat, MODIS EVI, and MCD12Q2 in the Chicago metropolitan area in representative years (a) and

the temporal trend at the regional level (b). Solid lines are the mean SOSs at the regional level and shadowed frames indicate the range of

SOS within the 25th and 75th quantile levels. Pixels without good fitting performance (i.e., the correlation coefficient is lower than 0.85)

5    were removed in the derived SOS from Landsat and MODIS EVI.



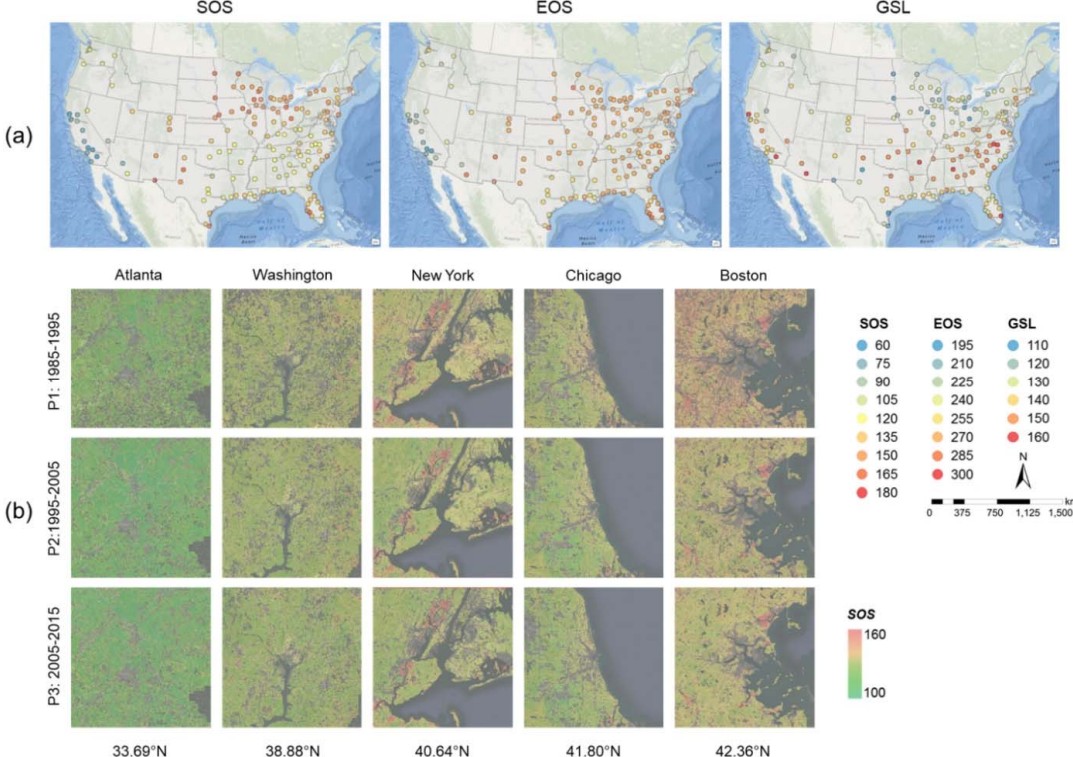

**Fig. 11:** Spatial patterns of the mean (1985-2015) vegetation phenology indicators (SOS, EOS, and GSL) in the US cities (a) and SOS in representative cities in the past three decades (b). Each dot in (a) represents the center of the urban cluster, and the spatial extent of selected cities in (b) is 25 km × 25 km.



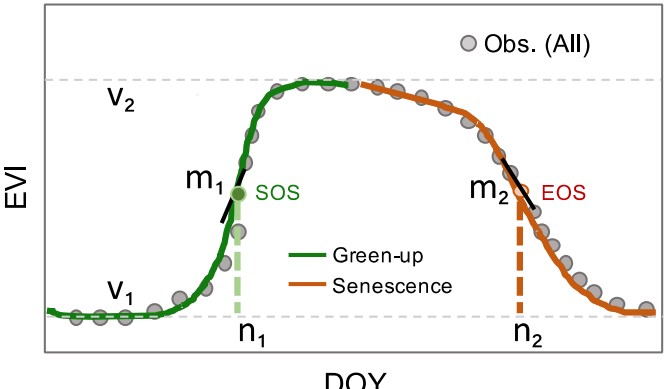

**Fig. A1:** Illustration of the double logistic model and corresponding parameters.