# Peer review of "A dataset of 30-meter annual vegetation phenology indicators (1985-2015) in urban areas of the conterminous United States"

_Earth System Science Data, 2019_

## Short Comment (SC1) · 3 Apr 2019

Amazing work. Do you mind to publish the corresponding GEE script?

---

## Referee Comment (RC1) · Francisco J Escobedo (Referee) · 8 Apr 2019

MS No. essd-2019-9A: dataset of 30-meter annual vegetation phenology indicators (1985-2015) in urban areas of the conterminous United States

Below is my review for the above manuscript. Overall I found it novel and a contribution to future research on the ecology of urban and peri-urban ecosystems in the United States. I have included some grammar and context suggestions as well as other suggestions. Key among these is that this data set can have many more applications that the ones currently presented in the manuscript. I have included some of these suggestions.

[Figure]

In terms of data quality, I did view some scenes directly on the Figshare site and did notice some problems with the overall visual quality of the scenes as many were distorted and banded. I also downloaded and opened in ArcCatalog and Arcmap some of the raster datasets. I did have a difficult time in trying to determine the overall content and geographic location of many of the images. The SHP file did help but a summary and description of each raster data set might have been helpful. However, I did not open every single raster data set or overlay some of these onto other geospatial data to assess their quality.

Below are some comments regarding the manuscript which I hope you will find useful.

Abstract Page (P) 2, Line (L) 10 and P3, L2: I would think that currently "fine-resolution" refers to submeter imagery. Landsat may have been fine in the 1980-90s but now it is considered medium resolution I would think.

P2, L 10-14: Atmospheric, soil and light pollution in urban environments will also drive plant phenology (e.g., leaf deciduousness-senescence), such a dataset can have applications to address issues like these.

P2, L 15: Urban morphology might be more relevant than "development"

P3, L5, "There have been" few attempts... Also specify what you mean here by "large scale". I believe you mean something like "regional (as opposed to local) scale".

P3, L13, "...therefore lack large-scale..."

P3 L14, "...urban ecosystem are more complex..." Mention something about the high floral species richness found in cities, relative to rural area in these latitudes.

P4, L9: Perhaps say urban areas instead of cities.

P4L 10: You did not use Landsat for the NTL observation I imagine? P4 L 18-19: Specify all the correction procedures (e.g., cloud removal) used.

P6 L 1-5: As mentioned in the introduction, urban area are heterogeneous and complex, not only in terms of vegetation diversity and phenology, but land use/covers as well. How was this accounted for in your methods? That is "vegetation" and "urban areas" are not homogenous in terms of their spectral and environmental characteristics. Land cover change is mentioned in Line 5 but did you develop a new, or use an existing urban land cover classification?

P8 L 24-25: Was the fact that these appear exclusively to be deciduous forest types a coincidence? Specifically, how do evergreen forest types/trees affect the urban phenology results?

P10 L10-11: Can increased impervious surfaces, pollution or changing species composition over the analysis period, also be a correlate?

Conclusion: the first 2 paragraph are repetitive. Perhaps discuss some limitations and more applications e.g., many vegetation-air pollution deposition models need leaf on/off data, city-level urban tree cover classification need to be done during leaf on, etc.

Figure 1-11 A1. Please spell out all acronyms. The reader should not have to read the text to find what the acronyms and symbols are.

Data quality from: https://doi.org/10.6084/m9.figshare.7685645.v2 I viewed thumbnails of all the *.tif files but did not attempt to download all since I am having trouble with memory and the ArcGis license. I did download and view several files and viewed them in the figshare site. Some of these scenes in the figsahre.com site (e.g., US_uCluster_83_COR-1985-2015.tif) seemed to have distortion in the form of a distorted scene, specifically 1/3 of the image was banded and distorted. I did view several other raster data sets in ArcCatalog (e.g. The uCluster_USA_gt500) and a summary and descriptions in the "Description" or metadata of the Raster dataset would have been helpful in understanding the content data set. The uCluster_USA_gt500. However I am unable to fully assess data quality at this time.

---

## Referee Comment (RC2) · 21 Apr 2019

Li and coauthors produced a 30-meter phenology dataset using Google earth engine based Landsat images, and a double logistic model. The study is timely and important. The fine-resolution phenology dataset is valuable, and provide an avenue on the urban phenology study since its high importance in public health, i.e. pollen allergy diseases, as well as urban ecosystem response to future climate warming. I would thus like to recommend to publish this nice study in the ESSD. Below, please find some suggestions that I hope can be help to improve the MS.

first, the authors argued that the logistic model is valuable to capture the trends of

green-up and senescence by using the pair-parameters, but the description is weaker, please specify or update this. In addition, the half-maximum criterion was used to extract the sos and eos, but the more popular method is using the maximum change rate. Different methods might generate different results, see the figure 10, between the modis evi and MCD12Q2, and large difference was obtained. I do not say the half-maximum is wrong, but the authors should address this issue in the discussion, and remind the reader to cite the method when using the dataset.

second, the Landsat phenology dataset was compared with in situ phenology data, including both phenoCam and ground observations, and I found the authors overestimated the results, i.e. a good agreement between these datasets. See the figure 6 and 7, the difference between Landsat and phenoCam is even larger than 20 days, i.e. RMSEs, for both SOS and EOS. Actually, I do not expect a high agreement, due to the forest structure and the difference of scale between Landsat (30m) and in situ observations (500m for PhenoCam). So, I would suggest to update the descriptions of these comparisons, and highlight the scale issues between the Landsat and PhenoCam and ground dataset.

some minor comments,

page 5, line 15, the physical meaning of parameter is related to vegetation growth and senescnece, please specify;

page 6, line 10-15, why the half-maximum criterion is likely to produce the sos and eos when leaves are likely to emerge? remove this argument or update;

page 7, 4.1 section, the authors argued the urban-rural gradient, but in the following the forest was presented as example. better remove this gradient arguments;

page 8, the argument in line 6, i.e. a good agreement with in-situ phenocam results, is in conflict with the line 15, i.e. the agreement is relatively weaker.. please improve these arguments;

fig3, specify GLP in the legend;

[Figure]

---

## Author Comment (AC1) · 22 May 2019

We thank David Carlson and the two referees for their time and effort to review our manuscript, and for their very positive and constructive feedback, which helped to further increase the quality of the paper.

**1 Francisco J Escobedo (Referee)**

Comment #1-1

MS No. essd-2019-9: A dataset of 30-meter annual vegetation phenology indicators (1985-2015) in urban areas of the conterminous United States. Below is my review for

[Figure]

the above manuscript. Overall, I found it novel and a contribution to future research on the ecology of urban and peri-urban ecosystems in the United States. I have included some grammar and context suggestions as well as other suggestions. Key among these is that this data set can have many more applications that the ones currently presented in the manuscript. I have included some of these suggestions.

Response: thank you for your positive comments. We addressed your concerns and made corresponding revisions. Details can be found in our attached response letter.

Comment #1-2

In terms of data quality, I did view some scenes directly on the Figshare site and did notice some problems with the overall visual quality of the scenes as many were distorted and banded. I also downloaded and opened in ArcCatalog and ArcMap some of the raster datasets. I did have a difficult time in trying to determine the overall content and geographic location of many of the images. The SHP file did help but a summary and description of each raster data set might have been helpful. However, I did not open every single raster data set or overlay some of these onto other geospatial data to assess their quality.

Response: we appreciate your suggestions for our dataset. The visualization of thumbnails in FigShare is not accurate due to its small size. The distorted and banded regions do not exist when visualizing the data in professional software such as ArcMap and Matlab. We provided a detailed example to explain this issue. As suggested, we also added a field of "CityName" in our provided SHP file, which helps to locate the region of each scene quickly. For each scene, we offered a COR layer for the fitting performance of the double logistic model. This helps users understand the uncertainty of the derived phenology indicator at the pixel level. Accordingly, we updated our data descriptions in FigShare (https://doi.org/10.6084/m9.figshare.7685645). We provided a more detailed explanation of our dataset in the response to comment #1-17.

Comment #1-3

Abstract Page, (P) 2, Line (L) 10 and P3, L2: I would think that currently "fine-resolution" refers to sub meter imagery. Landsat may have been fine in the 1980-90s but now it is considered medium resolution I would think.

Response: agree. We revised it in our manuscript. Below are two examples.

"Medium-resolution satellite observations show great potential for characterizing seasonal and annual dynamics of vegetation phenology in urban domains" (page 1, line 10).

"The derived phenology product in the US urban domains at the national level is of great use for urban ecology studies for its medium spatial resolution (30 m) and long temporal span (30 years)" (page 2, line 1).

Comment #1-4

P2, L 10-14: Atmospheric, soil and light pollution in urban environments will also drive plant phenology (e.g., leaf deciduousness-senescence), such a dataset can have applications to address issues like these.

Response: thank you. We added potential applications of our dataset in urban environments as suggested.

"Changes in the urban environment due to atmospheric, soil, and light pollutions will affect plant phenology (e.g., leaf senescence) (Escobedo et al., 2011), resulting in different phenology characteristics in urban ecosystems" (page 2, line 14).

Escobedo, F.J., Kroeger, T., & Wagner, J.E. (2011). Urban forests and pollution mitigation: Analyzing ecosystem services and disservices. Environmental Pollution, 159, 2078-2087.

Comment #1-5

P2, L 15: Urban morphology might be more relevant than "development"

[Figure]

Response: done.

Comment #1-6

P3, L5, "There have been" few attempts. . . Also specify what you mean here by "large scale". I believe you mean something like "regional (as opposed to local) scale".

Response: yes. We clarified this sentence as below.

"There are few attempts of mapping vegetation phenology in urban domains using Landsat observations at a regional (or global) scale due to complex vegetation compositions in urban ecosystems and the large dataset required for analysis" (page 3, line 7).

Comment #1-7

P3, L13, ". . .therefore lack large-scale. . ."

Response: done.

Comment #1-8

P3 L14, ". . .urban ecosystem are more complex. . ." Mention something about the high floral species richness found in cities, relative to rural area in these latitudes.

Response: thank you. We discussed the difference of species richness in cities and rural areas.

"vegetation types and compositions in urban ecosystems are more complicated, and the floral species are more abundant in cities than surrounding rural areas (Luz de la Maza et al., 2002)" (page 3, line 16).

Luz de la Maza, C., Hernández, J., Bown, H., Rodríguez, M., & Escobedo, F. (2002). Vegetation diversity in the Santiago de Chile urban ecosystem. Arboricultural journal, 26, 347-357.

Comment #1-9

P4, L9: Perhaps say urban areas instead of cities.

Response: done.

Comment #1-10

P4L 10: You did not use Landsat for the NTL observation I imagine?

Response: Yes, we derived urban extents from NTL observations (Zhou et al., 2018).

Zhou, YY., Li, XC., Asrar, G, Smith, S, & Imhoff, M. (2018). A global record of annual urban dynamics (1992-2013) from nighttime lights. Remote Sensing of Environment, 219, 206-220.

Comment #1-11

P4 L 18-19: Specify all the correction procedures (e.g., cloud removal) used.

Response: thank you. We specified correction procedures used in Landsat images.

"The correction of atmospheric effect was performed using the Landsat ecosystem disturbance adaptive processing system (LEDAPS) (Masek et al., 2006), and clouds and shadows were removed using the function of mask procedure (Fmask) (Zhu and Woodcock, 2012) before compositing the EVI time series" (page 4, line 21).

Comment #1-12

P6 L 1-5: As mentioned in the introduction, urban area are heterogeneous and complex, in terms of not only vegetation diversity and phenology, but land use/covers as well. How was this accounted for in your methods? That is "vegetation" and "urban areas" are not homogenous in terms of their spectral and environmental characteristics. Land cover change is mentioned in Line 5 but did you develop a new, or use an existing urban land cover classification?

Response: thank you for your questions. We did not separately calculate phenology indicators before and after land use cover change because we used the long-term

Landsat time series data to derive the mean phenology pattern. This could introduce the uncertainty in our result in areas with land cover changes. The uncertainties caused by such changes can be detected in the fitting performance of the double logistic model. These changed pixels can be excluded for specific applications by users. We clarified this in our revised manuscript as follows:

"sigmoid curves during green-up and senescence phrases are notably different across different vegetation cover types (e.g., forest and cropland). We evaluated the performance of the fitted double logistic model based on the correlation between the fitted and observed EVI observations. Pixels with land use/cover change during the study period or weak vegetation signals (e.g., extremely high built-up area or barren land) could have a low fitting performance, and these pixels can be excluded for specific applications." (page 6, line 6).

Comment #1-13

P8 L 24-25: Was the fact that these appear exclusively to be deciduous forest types a coincidence? Specifically, how do evergreen forest types/trees affect the urban phenology results?

Response: thank you for the questions. Yes. The dominant vegetation type in Harvard forest is the deciduous forest. For other vegetation types like the evergreen forest, the performance of our derived phenology results depends on the ability of the remote sensing signal to capture their reflectance signature. As shown in Fig. 4 (site 1), we observed a clear phenology pattern in Site 1 (Florida), where the evergreen forest is the dominated vegetation type. For regions (e.g., grassland) with relatively low performance, they can be indicated in the associated uncertainty in our dataset. We further explained this in the revised manuscript.

"Three dominant species of deciduous forest in the HF, including the red oak (Quercus rubra; QURU), red maple (Acer rubrum; ACRU), and yellow birch (Betula alleghaniensis; BEAL), were used in our analysis. However, for other vegetation types (e.g., evergreen forest), discernible phenology patterns can be also captured using the proposed methodology (e.g. Fig. 4, Site 1)." (page 9, line 8).

Comment #1-14

P10 L10-11: Can increased impervious surfaces, pollution or changing species composition over the analysis period, also be a correlate?

Response: thank you for your question. We clarified it in our revised manuscript.

"In addition, changes in urban environment such as impervious surface, air pollution, and species compositions can affect the spatiotemporal pattern of vegetation phenology in urban ecosystems (Li et al., 2015; Escobedo et al., 2011)" (page 10, line 24).

Comment #1-15

Conclusion: the first two paragraphs are repetitive. Perhaps discuss some limitations and more applications e.g., many vegetation-air pollution deposition models need leaf on/off data, city-level urban tree cover classification need to be done during leaf on, etc.

Response: thank you for your suggestions. The first paragraph summarizes the main steps we used to obtain our results and the second one presents evaluations of our dataset in comparison with other products. Thus, we would prefer to keep the current form of these two paragraphs. As suggested, we added discussion of limitations and potential applications of our dataset in urban studies.

"In addition, the derived leaf on/off information in this dataset is potentially useful for vegetation-air pollution deposition models (Escobedo and Nowak, 2009). However, it is worth noting that this dataset is most applicable for deciduous forest type. For grassland and evergreen forests in tropical areas, the uncertainty could be high in the derived phenology indicators. In addition, our phenology algorithm did not specifically consider pixels with land cover changes, which could be further improved when the product of annual urban dynamics becomes available." (page 12, line 7).

Escobedo, F.J., & Nowak, D.J. (2009). Spatial heterogeneity and air pollution removal by an urban forest. Landscape and Urban Planning, 90, 102-110.

Comment #1-16

Figure 1-11 A1. Please spell out all acronyms. The reader should not have to read the text to find what the acronyms and symbols are.

Response: thank you. We spelled out all acronyms in the revised figures.

Comment #1-17

Data quality from: https://doi.org/10.6084/m9.figshare.7685645.v2. I viewed thumbnails of all the *.tif files but did not attempt to download all since I am having trouble with memory and the ArcGis license. I did download and view several files and viewed them in the figshare site. Some of these scenes in the figsahre.com site (e.g., US_uCluster_83_COR-1985-2015.tif) seemed to have distortion in the form of a distorted scene, specifically 1/3 of the image was banded and distorted. I did view several other raster data sets in ArcCatalog (e.g. The uCluster_USA_gt500) and a summary and descriptions in the "Description" or metadata of the Raster dataset would have been helpful in understanding the content data set. The uCluster_USA_gt500. However, I am unable to fully assess data quality at this time.

Response: we appreciate your efforts to examine our dataset and your suggestions. We noticed that FigShare has some problems in thumbnails. Below is an example of the image "US_uCluster_83_COR-1985-2015" in FigShare and ArcMap. These banded and distorted areas in FigShare does not exist in ArcMap. We also found similar issues in other scenes in FigShare thumbnails. We explained this in the data description for awareness of the users. We also added a new field of region/city name in the provided shapefile of uCluster_USA_gt500, to help locate the region of interest using the information of cityName.

/* Insert Fig. R1 */

Fig. R1. An illustration of COR images visualized in FigShare (left) and ArcMap (right).

**2 Yongshuo Fu (Referee)**

Comment #2-1

Li and coauthors produced a 30-meter phenology dataset using Google earth engine based Landsat images, and a double logistic model. The study is timely and important. The fine-resolution phenology dataset is valuable, and provide an avenue on the urban phenology study since its high importance in public health, i.e. pollen allergy diseases, as well as urban ecosystem response to future climate warming. I would thus like to recommend publishing this nice study in the ESSD. Below, please find some suggestions that I hope can be help to improve the MS.

Response: thank you for your positive feedback. Below please find our detailed response to each comment.

Comment #2-2

First, the authors argued that the logistic model is valuable to capture the trends of green-up and senescence by using the pair-parameters, but the description is weaker. Please specify or update this. In addition, the half-maximum criterion was used to extract the SOS and EOS, but the more popular method is using the maximum change rate. Different methods might generate different results, see the figure 10, between the MODIS EVI and MCD12Q2, and large difference was obtained. I do not say the half-maximum is wrong, but the authors should address this issue in the discussion, and remind the reader to cite the method when using the dataset.

Response: thank you for your suggestion. We clarified the double logistic model in the revised manuscript.

"That is, $n_1$ and $n_2$ are dates with the maximum increasing and decreasing rates of green-up and senescence in sigmoid curves, while $m_1$ and $m_2$ are slopes that determine the shape of sigmoid curves" (page 5, line 24).

The half-maximum criterion (i.e., middle of the sigmoid curve) determines the dates with the maximum increasing and decreasing rates of the first derivative (i.e., change rate) in EVI (Fisher et al., 2006). This method was applied to the Landsat and MODIS EVI data. While for the phenology product of MCD12Q2, SOS and EOS are the dates at the base of the sigmoid curve (i.e., the inflection point). Thus, the derived SOS using our method is later than MCD12Q2 while EOS is earlier than MCD12Q2. We clarified this in the revised manuscript.

"We derived phenology indicators of SOS and EOS using a half-maximum criterion method (Fisher et al., 2006). In this method, SOS and EOS were calculated as dates when the first derivative of EVI reaches the maximum increasing and decreasing rates during the green-up and senescence phases, respectively. Although there are other definitions of SOS and EOS such as inflection points (i.e., at the base of sigmoid curve) (Zhang et al., 2003), the criterion used in the study is more temporally stable and can be applied to plants with different canopy structures (Fisher and Mustard, 2007)" (page 6, line 14).

"It is worth noting that SOS derived from the half-maximum criterion in this study is consistently later compared to the MODIS product using the criterion of the inflection point" (page 10, Line 13).

Comment #2-3

Second, the Landsat phenology dataset was compared with in situ phenology data, including both phenoCam and ground observations, and I found the authors overestimated the results, i.e. a good agreement between these datasets. See the figure 6 and 7, the difference between Landsat and phenoCam is even larger than 20 days, i.e., RMSEs, for both SOS and EOS. Actually, I do not expect a high agreement, due to the forest structure and the difference of scale between Landsat (30m) and in situ observations (500m for PhenoCam). Therefore, I would suggest updating the descriptions of these comparisons, and highlighting the scale issues between the Landsat and

PhenoCam and ground dataset.

Response: thank you for your suggestion. We agreed that the difference between Landsat and in-suit derived phenology results is mainly attributed to the scale effect of these two datasets, as well as the difference in background vegetation types and used vegetation indices. We revised our description of data comparison (i.e., removing phrases like "good agreement") and discussed these factors in our revised text.

"For those sites (blue or light blue dots) with large differences, the performance of fitting Landsat EVIs using the double logistic model is relatively low because these sites are mainly embedded in ecosystems that are dominated by shrubs, evergreen forests, or wetlands (Fig. S3)" (page 8, line 4).

"Discrepancies between these two sets of phenology indicators derived from Landsat and PhenoCam are mainly attributed to factors such as: (1) two different vegetation indices (i.e., EVI and GCC); and (2) the scale effect between in-situ PhenoCam and Landsat observations (Liu et al., 2017)" (page 8, line9).

"Overall, the derived phenology indicators (SOS and EOS) are spatially consistent with those from in-situ PhenoCam data at the national scale (Fig. 6)" (page 7, line 21).

Comment #2-4

Page 5, line 15, the physical meaning of parameter is related to vegetation growth and senescence, please specify;

Response: thank you for your suggestion. We clarified these parameters in our revised manuscript.

"where $f(t)$ is the fitted EVI value at the day $t$; $v_1$ and $v_2$ are the background and amplitude of EVI over the entire year, respectively; and $m_1$ & $n_1$, $m_2$ & $n_2$ are the pair-parameters that capture the trend of green-up and senescence phases of vegetation growth. That is, $n_1$ and $n_2$ are dates with the maximum increasing and decreasing rates of green-up and senescence sigmoid curves, and $m_1$ and $m_2$ are

the slopes that determine the shape of two sigmoid curves" (page 12, line 17).

Comment #2-5

Page 6, line 10-15, why the half-maximum criterion is likely to produce the SOS and EOS when leaves are likely to emerge? Remove this argument or update.

Response: we removed this sentence in the revised manuscript as suggested.

Comment #2-6

Page 7, 4.1 section, the authors argued the urban-rural gradient, but in the following the forest was presented as example. Better remove this gradient arguments.

Response: thank you for your suggestion. Forest is an example for illustrating latitudinal difference of vegetation phenology in the eastern United States (Fig. 4), while for the method performance in urban ecosystems, we selected different sites (with different vegetation covers) from urban core to rural areas for illustration (Fig. 5). As suggested, we removed "urban-rural gradient".

"The performance of the developed GEE-based double logistic model is reasonably good across different latitudes and different vegetation types in urban ecosystems" (page 7, line 12).

"At the city scale, the proposed double logistic model shows a good performance of fitting EVI time series from urban to rural areas (Fig. 5)" (page 7, line 15).

Comment #2-7

Page 8, the argument in line 6, i.e. a good agreement with in-situ phenocam results, is in conflict with the line 15, i.e. the agreement is relatively weaker. Please improve these arguments.

Response: thank you for your suggestion. We clarified it in the revised manuscript.

"Overall, the annual variability of SOS derived from Landsat observations is consistent

with that from the in-situ PhenoCam observations (Fig. 7)" (page 8, line 12).

Comment #2-8

Fig3, specify GLP in the legend;

Response: GLP is the abbreviation of "generalized Landsat phenology". We clarified it in the figure caption.

"Fig. 3. Illustration of the generalized Landsat phenology (GLP) approach for identifying the annual variability of phenology indicators."

**3 Dongdond Kong**

Comment #3-1

Amazing work. Do you mind to publish the corresponding GEE script?

Response: thank you. We are pleased to share upon request the script for research and education by the academic community.

Please also note the supplement to this comment:
https://www.earth-syst-sci-data-discuss.net/essd-2019-9/essd-2019-9-AC1-supplement.pdf

**COR**

0

**Thumbnail in FigShare**          **COR Map**

**Fig. 1.** Fig. R1. An illustration of COR images visualized in FigShare (left) and ArcMap (right).

---

## Author Comment (AC2) · 22 May 2019

Dear Reviewers,

Thank you very much for your comments and suggestions which are very helpful to improve the manuscript. We also provided revised manuscripts (clean and tracked) for your references.

Best wishes,

Yuyu Zhou

[Figure]

Please also note the supplement to this comment:
https://www.earth-syst-sci-data-discuss.net/essd-2019-9/essd-2019-9-AC2-
supplement.zip

---

## Author Response (AR2)

**#1 Tropical Editor**

**Comment #1-1**

Please apply a consistent language for changes in SOS and EOS. Although the magnitude of the DOY values increase or decrease, from a phenological point of view the SOS dates advance or occur earlier while the dates for EOS retard or delay or occur later. In effect, the growing season extends (GSL increases). In places, e.g. on page 8, you use the mathematical terms 'decrease' or 'increase' while on pages 10 and 12 you use the term 'advancement' or 'advanced'. The phenological community must have standard terms for these changes? You will help potential users by adopting better and more consistent descriptive terms.

**Response:** thank you for your suggestion. Yes, we agree we should use consistent terms of "advanced" or "earlier" to describe changes of phenology indicators. As suggested, we carefully checked these terms we used in our manuscript, for a more consistent description of phenology change (e.g., advanced or delayed).

 *"A detailed illustration of the Acadia site indicates the SOS derived from the two datasets is notably **advanced** during period 2006-2010 and their corresponding EOS is **delayed** after 2011. Although magnitudes of SOS and EOS are different over the years, their temporal trends (i.e., **advanced** or **delayed**) are relatively consistent."(Page 8)*

**Comment #1-2**

In one case (e.g., top of page 10 and Figure 9), you describe a magnitude of SOS from MODIS exceeding 180 days. You highlight that obviously too-high value as a likely error, but give us no explanations? In figure 11 you only allow a max value for your Landsat-derived SOS of 160?

**Response:** thank you for your questions. In general, for most vegetation types, expecting for agriculture land (due to human management) and grass land (sensitive to precipitation), the onset of green-up (i.e., SOS) is likely to occur in late April (~ 100) to earlier June (~ 160) (*Piao et al., 2006*). Commonly, pixels with SOS later than 180 were excluded due to the high degree of uncertainty (*Zhou et al., 2016*).

In our data, we did not exclude these extreme values. We only visualized our data in the paper with maximum values of SOS constrained in the legend for better comparisons. In Fig. 9, we set the maximum value of SOS in the legend as 180 to have a better comparison with MOD12Q2 product, which has the maximum value over 180. In Fig. 11, we set the maximum value of SOS as 160 in the legend for a better cross-region comparison (a narrow dynamic range) of our SOS data. For clarity, we revised our description as below.

 *"Also, there might be uncertainties in MCD12Q2 in highly urbanized regions with SOS above 180 days (Zhou et al., 2016) (Fig. 9a)" (page 10)*

➢ *Zhou, Decheng, Shuqing Zhao, Liangxia Zhang, and Shuguang Liu. "Remotely sensed assessment of urbanization effects on vegetation phenology in China's 32 major cities." Remote Sensing of Environment 176 (2016): 272-281.*
➢ *Piao, Shilong, Jingyun Fang, Liming Zhou, Philippe Ciais, and Biao Zhu. "Variations in satellite‐derived phenology in China's temperate vegetation." Global change biology 12, no. 4 (2006): 672-685.*

**Comment #1-3**

With reference to Section 4.5 on page 10, Figure 11 shows spatial patterns (e.g. 11a) but the temporal patterns (e.g. from P1 to P2 to P3) would only appear in 11b and, in any case, seem very weak at least in this presentation. Perhaps in Boston but not in the other four locations? One does not observe temporal

trends in Figure 8 or Figure 10; if statistically-valid temporal trends exist you have not pointed them out? Therefore, where in the data would a user actually find the imagery to address these questions of advanced SOS or extended GSL? Give us an example?

**Response:** thank you for your questions and suggestions. For Fig. 11(b), we added the mean SOS of each city (bolded text) and its standard deviation (in parentheses) within each period. Although the SOS from P1 to P3 is not significantly advanced, overall the SOS in P3 (2005-2015) is earlier than that in P1 (1985-1995). In addition, another purpose of Fig. 11(b) here is to provide illustrative examples of SOS change from the South to North, as a response to Fig. 11(a). Therefore, for clarity, in our revised manuscript, we revised our description about temporal change of SOS.

[revised manuscript text omitted]